# High-Sensitivity Cardiac Troponin Impact on the Differential Diagnosis of Non-ST Segment Elevation Coronary Syndromes—Is It Helping?

**DOI:** 10.3390/medicina58081084

**Published:** 2022-08-11

**Authors:** Kristina Šulskutė, Aistė Pilkienė, Emilija Meškėnė, Džiugilė Kersnauskaitė, Rokas Šerpytis, Žaneta Petrulionienė, Pranas Šerpytis

**Affiliations:** 1Department of Cardiovascular Medicine, Vilnius University, 01513 Vilnius, Lithuania; 2Centre of Cardiology and Angiology, Vilnius University Hospital Santaros Klinikos, 08410 Vilnius, Lithuania

**Keywords:** guidelines, high sensitivity troponin, NSTEMI, requalification, unstable angina

## Abstract

*Background and Objectives*: Increased levels of high-sensitivity cardiac troponin (hs-cTn) are the main criteria that differentiate non-ST segment elevation myocardial infarction (NSTEMI) from unstable angina (UA). How are these implemented in clinical practices? This study aims to detect cases of misdiagnosed UA instead of NSTEMI. *Materials and Methods*: We analysed discharge summaries of 840 patients admitted to Vilnius University Hospital Santaros Klinikos with the diagnosis of UA in 2017–2018. We retrospectively checked symptoms, levels of hs-cTn, coronary angiography and electrocardiogram changes with an aim to differentiate UA and type 1 NSTEMI, according to the Fourth Universal Definition of Myocardial Infarction. We excluded patients with missing hs-cTn levels or coronary angiography. *Results*: We found that 46.71% (*n* = 334) of patients met the diagnostic criteria of UA according to the Fourth Universal Definition, whereas 19.16% of patients (*n* = 137) could have been diagnosed with type 1 NSTEMI instead of UA. In the group of patients who could be reclassified to type 1 NSTEMI, the median level of hs-cTn was 184.32 [226.15] ng/L on admission. The median of the lowest level during the hospitalization was 114.0 [207.4] ng/L. Median highest—304.0 [257.6] ng/L. Myocardial infarction with non-obstructive coronary arteries could have been diagnosed in 3.36% (*n* = 24) of patients. *Conclusions*: Only less than half of patients met the diagnostic UA criteria. Almost one-fifth of patients with a diagnosis of UA could be reclassified to type 1 NSTEMI.

## 1. Introduction

Central and Eastern European regions are known to have the highest cardiovascular disease burden worldwide, mainly because of coronary artery disease (CAD) [1,2,3]. Unstable angina (UA) is diagnosed in approximately 8.9–10% of patients complaining of chest pain when there is no evidence of acute cardiomyocyte necrosis [4,5]. An increased level of troponin is the main criterion that differentiates myocardial infarction without elevation of the ST segment (NSTEMI) from the UA. Type 1 NSTEMI is diagnosed when cardiac troponin is increased by at least one value above the 99th percentile upper reference limit, and is accompanied by one or more other clinical signs. These signs are symptoms of acute myocardial ischemia, new ischaemic electrocardiogram (ECG) changes without ST-segment elevation, pathological imaging evidence and/or angiography which shows a coronary thrombus [6].

In the era of high-sensitivity cardiac troponin (hs-cTn) assay, UA diagnosis should decrease; meanwhile, NSTEMI incidence should increase [7]. Accurate diagnosis is highly important for both health care professionals and patients: it helps to select optimal treatment and benefits for further drug reimbursement and rehabilitation. Vilnius University Hospital Santaros Klinikos Centre of Cardiology and Angiology is a tertiary care center. A broad spectrum of procedures are available for managing patients with acute coronary syndromes. The center performs about 2500 percutaneous coronary interventions (PCI). Overall, 700 of them are primary PCI, and about 900 cases of MI are hospitalized. All of the procedures performed are free of charge for the patients since we have universal health care provided by the state. In this study, we seek to evaluate the diagnosis of UA in the Centre of Cardiology and Angiology, and to determine the potential to reclassify patients from UA to NSTEMI, according to the Fourth Universal Definition of Myocardial Infarction.

## 2. Materials and Methods

We conducted a retrospective single-center analysis of discharge summaries of 840 patients with UA diagnosis who were hospitalized in Vilnius University Hospital Santaros Klinikos Centre of Cardiology and Angiology during the period of 2017–2018. Patients who have not been tested for a high-sensitivity cardiac troponin I (hs-cTnI) and have not undergone coronary angiography (CAG) were excluded (Figure 1). The final research sample included 715 patients. The normal range of hs-cTnI was referred to as ≤15.6 ng/L in women and ≤35.2 ng/L in men. We used the data of hs-cTnI, which were taken on admission (in the emergency room), and the smallest and largest values during the time of hospitalization. Inverted T, ST-segment depression, and newly found left bundle branch block were considered to be ischaemic ECG changes. 

For the threshold, we chose a > 5-times increase in the hs-cTnI. It is because elevations up to five-fold beyond the upper reference limit have high (>90%) positive predictive values (PPV) for acute type 1 myocardial infarction (MI), whereas elevations up to three-fold greater than the upper reference limit have only limited (50–60%) PPV for acute MI, and may be associated with a broad spectrum of conditions (e.g., tachyarrhythmias, myocarditis, heart failure) [5]. 

We held that patients were correctly diagnosed with UA if they had normal or <3-times increased hs-cTnI, and met one or more of the following criteria: Rest angina pectoris that is prolonged (usually >20 min);New-onset angina of at least class 3 severity in the Canadian Cardiovascular Society classification;Destabilization of previously stable angina.

UA was reclassified to myocardial infarction in cases where hs-cTnI was increased ≥5 times: Type 1 NSTEMI if followed by obstructive CAD (≥50% diameter stenosis in a major epicardial vessel);Myocardial infarction with non-obstructive coronary artery (MINOCA) in cases with no angiographic obstructive coronary artery disease.

We considered patients as “probable” MI or “probable” MINOCA if they met the criteria of increased hs-cTnI ≥3 but <5 times.

After reclassification, we did not include MINOCA and “probable” MINOCA groups in the final statistical analysis due to the small sample. For final statistical analysis, we chose to compare the UA and NSTEMI groups because of a large sample, and these groups best met reclassification criteria; there is no need for further investigations for these diagnoses to be confirmed.

We used MS Excel, R-Commander, and SPSS21 for statistical analysis. *p*-value <0.05 indicates significant findings. To determine if a data set is in a normal distribution, we used a Kolmogorov–Smirnov test. Mean and standard deviation present normally distributed data. The median and interquartile ranges present not-normally distributed data. For numerical data comparisons, we used a Mann–Whitney U test. To compare proportions between UN and type 1 NSTEMI, we applied Chi-squared and Fisher’s exact tests. The Local Ethics Committee of Vilnius University approved the study protocol (consent Nr.158200-18/4-1015-522, 2018-04-03). 

## 3. Results

### 3.1. Baseline Characteristcs

Out of 715 analysed patients, 458 (64.1%) were men (median age (interquartile range) 66 (17) years) and 257 (35.9%) were women (median age 72 (14) years). One-third of patients (237 (33.1%)) had history of previous myocardial infarctions (MI). At the time of hospital admission, 615 (86.0%) patients had chest pain. Out of these 615 patients, 362 (50.6%) said that pain started while undergoing some sort of physical activity. According to patients’ anamnesis, 529 (74%) felt similar chest pain more than once. We identified that 133 (18.6%) patients felt chest pain for longer than 20 min. Detailed data are presented in Table 1.

We analysed ECGs of all patients: 336 (46.99%) had no ischaemic changes, whereas 364 (50.91%) had some of the ischaemic changes (see methods). The most common ischaemic changes were the inverted T wave in 206 patients (28.81%) and the ST segment’s depression in 133 (18.6%) cases (Table 1). 

We also checked CAG protocols and found that 608 (85.0%) patients had obstructive CAD. The left anterior descending artery (LAD) had ≥50% diameter stenosis in 419 (64.86%) cases, right coronary artery (RCA)—301 (46.52%) cases. The left circumflex artery (LCX)—291 (45.05%), and left main stem—53 (8.2%) cases.

### 3.2. Diagnostic Reclassification by a High-Sensitivity Cardiac Troponin I Assay

We found that hs-cTnI levels were increased by 5 or more times in 118 (17%) cases (measured on admission in emergency department) (Figure 2). 

We followed recommendations of the Fourth Universal Definition of Myocardial Infarction, and according to elevated hs-cTnI levels and changes of CAG, 137 (19.16%) patients could have been diagnosed with type 1 NSTEMI. Median level of hs-cTnI on admission for these patients was 184.32 [226.15] ng/L. When the patients were admitted, the median of the lowest hs-cTnI during time of hospitalization was 114.0 [207.4] ng/L, and the median of the highest was 304.0 [257.6] ng/L (Table 2). 

The median level of hs-cTnI for patients in the “probable” type 1 NSTEMI was 47.15 [67.25] ng/L on admission. The median of the lowest hs-cTnI level during hospitalization was 37.0 [50.8] ng/L and the median of the highest hs-cTnI—61.2 [55.2] ng/L (Table 2).

There were 24 (3.36%) patients who could have been diagnosed with MINOCA. They median hs-cTnI on admission was 18.3 [81.75]). We also found “probable” MINOCA for 36 (5.03%) patients, and their median hs-cTnI on admission was 12.85 [56.05]).

### 3.3. Unstable Angina Compared to Reclassified Myocardial Infarction

According to requalification criteria, we divided patients into two groups: the genuine UA group (a total of 334 patients), and type 1 NSTEMI (a total of 137 patients). We excluded patients with “probable” type 1 NSTEMI and MINOCA.

There were significantly more men in the UA group than women (227 (68%) vs. 107 (32%), *p* = 0.005). In the type 1 NSTEMI group, there was no significant difference between sexes (Table 1). Patients with type 1 NSTEMI were older than UA patients (71.41 vs. 65.59 years old, *p* < 0.001) (Table 1). 

Patients in the type 1 NSTEMI group experienced pain longer than 20 min more often compared with those in the UA group (31 patients (64.6%) vs. 54 (40.0%), *p* = 0.003). We found that patients in the UA group had previous episodes of pain (especially in the chest) more than those in the type 1 NSTEMI group (276 (85.7%) vs. 79 (57.7%), *p* < 0.001). The type of pain, irradiation site, history of MI, and frequency of coronary interventions did not differ between type 1 NSTEMI and UA groups (Table 1). However, we found that the type 1 NSTEMI group had ischemic ECG changes more often compared with the UA group (75 (54.7%) vs. 148 (44.3%), *p* = 0.039) (Table 1). 

## 4. Discussion

### 4.1. Unstable Angina Versus Myocardial Infarction

The Fourth Universal Definition of Myocardial Infarction suggests using hs-cTn in diagnosing acute coronary syndrome (ACS), leading to a decrease in the troponin diagnostic threshold for MI [6]. While it was thought that these new changes would decrease the rate of UA diagnosis and increase NSTEMI [8,9,10,11], there are studies suggesting the opposite—the diagnosis of UA rates increased in the new hs-cTn era [8,12]. We aimed to investigate how this new definition is implemented in clinical practice, since hs-cTnI was introduced to everyday clinical practice in 2017. Although guidelines establish that the 99th percentile of hs-cTn increase should be considered as MI, there is no unanimous threshold level of hs-cTn to diagnose MI in everyday clinical practice [5]. Some studies suggest a threshold of increase by 3 times, and others by 5 times [13]. In comparison, TOTAL-AMI registry made a similar analysis of patients with UA diagnosis and the threshold was >118 ng/L [12].

We considered the level of hs-cTnI increase more than 5 times to be the diagnostic criteria for diagnosing NSTEMI and results were alarming: almost every fifth patient had hs-cTnI increased by more than 5 times followed by ischemic changes in ECG, typical symptoms or obstructive CAD confirmed by CAG. These findings lead us to the conclusion that every fifth patient should be reclassified to the type 1 NSTEMI group instead of UA. Findings suggest that clinicians still do not rely on the suggested 99th hs-cTnI percentile (based on sex), and the UA is still being diagnosed incorrectly. Further work on consensus is still needed to establish thresholds of cardiac troponin used to diagnose patients with UA or NSTEMI.

Patients who were misdiagnosed with UA in our Clinics did not receive a full treatment: CAG and PCI were delayed, and no rehabilitation or further follow-up was provided. Our study still needs further investigation on patients’ follow-up, even though studies show that the Fourth Universal Myocardial Infarction Definition did not improve patients’ outcomes [14]. We suggest that additional studies should be carried out to investigate not only primary major endpoints but also to take rehabilitation and its benefits, life quality, and other aspects.

Worldwide data suggest that in the era of hs-cTn, the sensitivity of this laboratory test decreases. Therefore, many patients are misdiagnosed with ACS, which leads to excessive treatment including medical interventions [8]. Our study suggests the opposite: every fifth patient has been underdiagnosed, and thus does not receiving adequate treatment. There are studies stating that hs-cTn increases by more than 3 but less than 5 times may be explained by other conditions, and those patients should not be diagnosed with NSTEMI [13]. Our study took a closer look at this group of patients. Research has revealed that every fourth patient had an increase in cardiac troponin level from 3 to 5 times. Further analysis showed that all of them could be reclassified to the NSTEMI group due to having ischemic ECG changes, symptoms, and obstructive CAD confirmed by CAG. We suggest that the group of patients with an hs-cTn increase by 3 to 5 times should be taken into care and investigated more closely to prevent underdiagnosing and undertreatment. This is particularly important because a nation-wide SWEDEHEART study showed that hs-cTnI level > 30 ng/L is the cut-off point when invasive strategy improves outcomes [11].

### 4.2. The Difference in Baseline Characteristics

Previous studies have shown that the prevalence of UA is higher among men than among women, and that in the group of those diagnosed with UA, men are younger than women [15,16]. After reclassification according to hs-cTn value, it was observed that patients with genuine UA differed from those who were re-diagnosed with NSTEMI. The former group tended to be younger and had a lower prevalence of obstructive CAD, but there was no difference in incidence rate between sexes. It means that clinicians tend to underdiagnose women with NSTEMI. In this respect, this study has revealed similar findings as to the SWEDEHEART-registry-based analysis [16].

Although cigarette smoking is a traditional risk factor for CAD [17], this study revealed that only 25% of patients had records about smoking status taken. It suggests that not enough attention is paid to one of the most important avoidable causes of cardiovascular diseases [18].

### 4.3. The Difference in Clinical Characteristics

The main clinical presentation of ACS in patients presenting without persistent ST-segment elevation is acute chest discomfort [5]. Our study showed that not all patients complained about chest pain: some had atypical epigastric, interscapular, or left arm pain (Table 1). Almost 2.5% of cases did not have any records about pain.

While comparing patients who were re-diagnosed with type 1 NSTEMI, we found that patients with genuine UA more frequently had previous chest pain (before admission to the hospital). The 2020 ESC NSTEMI guideline proclaimed that atypical presentations including isolated epigastric pain are more often observed in older patients [5,19]. It could explain why non-chest pain more often occurs in patients with type 1 NSTEMI rather than in UA. However, prolonged pain (>20 min) was detected significantly more often in the type 1 NSTEMI group.

Trying to reach scientific information about UA’s clinical presentation, authors have encountered unexpected difficulties. Some researchers deliberately eliminated UA patients and just focused on NSTEMI or STEMI [20], or the UA group was not considered as an autonomous unit [21]. If researchers would address the diagnostic challenges of UA, it would be possible to develop further studies and offer updated information on this topic.

### 4.4. The Difference in Instrumental Diagnostics

ECG at presentation is a useful tool for risk prediction in ACS [5]. However, ECG patterns, their true frequency, and diagnostic yield are not widely analyzed in patients with ACS without persistent ST-segment elevation [5]. Our study revealed that half of ECGs showed ischemic changes including negative T wave, ST depression, and newly detected conductive disorders. Worse outcomes are expected for patients with ST-segment depression on ECG rather than with normal ECG [22]. 

According to CAG findings, LAD had significant stenosis the most frequently. LAD more often had been affected in the type 1 NSTEMI than the UA. Even though our study did not exclude patients with total occluded vessels on CAG, another study showed that one-quarter of patients presenting with NSTE-ACS with total occlusions are associated with increased mortality (with increasing frequency from the RCA, to the LCX, and to LAD) [23]. It could be predicted that our population has an increased mortality risk, even so, it is essential for future studies to confirm the hypothesis. 

Despite the decrease in UA diagnosis worldwide, clinicians tend to use UA as a “garbage bag” diagnosis. Our research revealed that about 15% of patients with discharge UA diagnosis had no hs-cTnI assay or CAG performed. That was due to various reasons, e.g., patients did not give their consent for CAG; there were no coronary signs of myocardial ischemia in non-invasive exercise tolerance test; CAG and PCI was performed recently; triple vessel disease was found and CAG at that moment was too risky. Fewer than 3% of cases had CT coronary angiography or exercise tolerance tests carried out. Better-defined management routines could help detect earlier signs of CAD and decrease false-positive UA diagnosis. Additionally, prior detection and proper medical or invasive strategy could prevent ACS and unnecessary hospitalization, and reduce the occupation of emergency departments. 

This study has four main limitations. First, the study is held only in one center, although there are six PCI centers in Lithuania, and each has its own practice and algorithms for diagnosing and managing UA and the type 1 NSTEMI. Second, we did not analyze MINOCA or probable MINOCA groups because of the small number of patients and the need for further investigations for these diagnoses to be confirmed. Third, the anamnesis data collected (i.e., chest pain characteristics, smoking anamnesis) are not very accurate because our center does not have one questionnaire for patients that arrive at the ER with chest pain, and data differ between each specialist. The last one is that we have not performed a follow-up of the UA and the type 1 NSTEMI group patients yet.

## 5. Conclusions

Only less than one-half of patients were correctly diagnosed with UA, with significantly more men than women. Almost one-fifth of patients with a diagnosis of UA could have been reclassified to type 1 NSTEMI. UA patients had previous episodes of pain more often than those who should be retained to type 1 NSTEMI diagnosis. Every fourth patient could have been classified as a “probable” type 1 NSTEMI patient. These findings show that MI is still underdiagnosed.

## Figures and Tables

**Figure 1 medicina-58-01084-f001:**
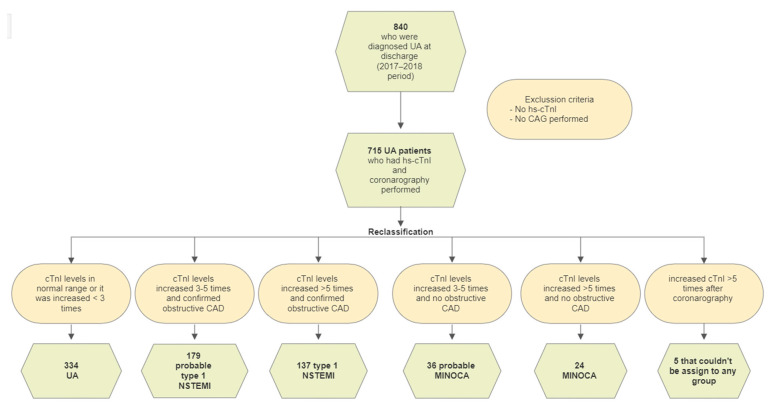
Flowchart of the patient selection process and reclassification results. CAG—coronary angiography; hs-cTnI—high-sensitivity cardiac troponin-I; MINOCA—myocardial infarction with non-obstructive coronary arteries; NSTEMI—non-ST segment elevation myocardial infarction; UA—unstable angina.

**Figure 2 medicina-58-01084-f002:**
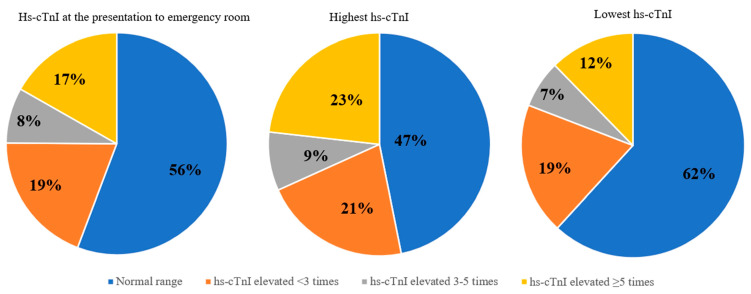
The proportion of normal range and elevated high sensitivity cardiac troponin-I values at the presentation to the emergency room, the highest and the lowest hs-cTnI values during hospitalization. hs-cTnI—high sensitivity cardiac troponin-I.

**Table 1 medicina-58-01084-t001:** Baseline demographic, clinical characteristics, instrumental diagnostics, and high sensitivity cardiac troponin-I of all patients and after reclassification with UA and type 1.

		Initial Diagnosis UA	After Reclassification		
		All (*n* = 715)	Type 1 NSTEMI (*n* = 137)	UA (*n* = 334)	*p* Value
**Baseline characteristics**					
**Age** (**years**)**, mean** (**SD**)		67.6 (11.02)	71.4 (9.68)	65.6 (11.02)	**<0.001**
**Age by sex** (**years**)**, median** (**min-max**)	Male	66 (35–91) ^a^	70 (44–90) ^b^	64 (36–91) ^c^	**<0.001**
	Female	72 (43–90) ^a^	74 (49–88) ^b^	68 (46–90) ^c^	**0.004**
**Sex, n** (**%**)	Male	458 (64.1) ^a^	86 (55.5)	227 (68.0)	0.01
	Female	257 (35.9) ^a^	61 (44.5)	107 (32.0)	
**History of previous MI1, n** (**%**)		237 (33.1)	47 (34.3)	113 (33.8)	0.92
**History of PCI, n** (**%**)		259 (36.2)	57 (41.6)	127 (38.0)	0.47
**History of CABG surgery, n** (**%**)		79 (11.0)	20 (14.6)	34 (10.2)	0.17
**Obstructive CAD, n** (**%**)		608 (58.0)	137 (100)	276 (82.6)	**<0.001**
**Smoking status, n** (**%**)	Smoking	111 (15.5)	13 (9.5)	54 (16.2)	0.13
	Unknown	538 (75.2)	113 (82.5)	248 (74.3)	
**Clinical characteristics**					
**Prolonged pain** (**>20 min.**)**, n** (**%**)		133 (18.6)	31 (64.6)	54 (40.0)	**<0.001**
**Previous chest pain, n** (**%**)		529 (74.0)	76 (57.7)	276 (85.7)	**<0.001**
**Exercise-induced chest pain, n** (**%**)		362 (50.6)	70 (51.5)	170 (52.8)	0.78
**Pain localisation, n** (**%**)	Chest	615 (86.0)	115 (83.9)	309 (92.5)	**<0.001**
	Epigastric region	12 (1.7)	1 (0.7)	8 (2.4)	
	Chest and Epigastric region	10 (1.4)	1 (0.7)	2 (0.6)	
	No pain at the presentation to emergency room	58 (8.1)	18 (13.1)	1 (0.3)	
	Intrascapular	4 (0.6)	1 (0.7)	2 (0.6)	
	Left arm	1 (0.1)	1 (0.7)	0	
**Felt typical pain** (**pressing, dull, tearing**)**, n** (**%**)		374 (52.3)	71 (51.8)	192 (57.5)	0.08
**Typical irradiation of pain** (**arm, neck, wide irradiation, mandible, epigastric region and no irradiation**)**, n** (**%**)		665 (93.0)	131 (95.6)	305 (91.3)	0.17
**Dyspnea, n** (**%**)		267 (37.3)	53 (39.0)	115 (35.3)	0.45
**Instrumental diagnostics**					
**Ischemic changes in ECG** (**inverted T, ST segment depression and new left bundle branch block**)		364 (50.9)	75 (54.7)	148 (44.3)	**0.04**
**ECG changes, n** (**%**)	No changes in ECG	336 (47.0)	61 (44.5)	176 (52.7)	0.2
	Negative T wave	206 (28.8)	35 (25.5)	89 (26.6)	
	ST depression	133 (18.6)	33 (24.1)	52 (15.6)	
	Left bundle branch block	25 (3.5)	6 (4.4)	10 (3.0)	
	Right bundle branch block	6 (0.8)	1 (0.7)	4 (1.2)	
	Left anterior fascicular block	5 (0.7)	1 (0.7)	0	
	Non specific ECG changes	4 (0.6)	0	3 (0.9)	
**PCI performed, n** (**%**)		354 (49.5)	76 (57.7)	154 (46.1)	**0.02**
**Findings of CAG**	LMS	53 (8.2)	16 (11.7)	22 (6.6)	0.07
	LAD	419 (64.7)	123 (89.8)	203 (60.8)	**<0.001**
	LCX	301 (46.5)	85 (62.0)	152 (45.5)	**0.001**
	RCA	291 (45.1)	83 (60.6)	145 (43.4)	**0.001**
**CABG surgery performed, n** (**%**)		42 (5.9)	10 (7.3)	14 (2.3)	0.17
**CT coronary angiography performed, n** (**%**)		20 (2.8)	1 (0.7)	10 (3.0)	0.19
**Exercise tolerance test****performed, n** (**%**)		19 (2.7)	2 (1.5)	14 (4.2)	0.17
**High sensitivity cardiac troponin-I (hs-cTnI)**					
**Hs-cTnI at the presentation to emergency room** (**ng/L**)**, median** (**IQR**)		17.6 (76.55)	184.4 (226.15)	6.4 (8.10)	**<0.001**
**Lowest hs-cTnI** (**ng/L**)**, median** (**IQR**)		14.1 (46.45)	114.0 (207.40)	5.7 (7.28)	**<0.001**
**Highest hs-cTnI** (**ng/L**)**, median** (**IQR**)		32.0 (107.85)	304.0 (357.60)	7.1 (8.80)	**<0.001**

IQR—interquartile range; CAD—coronary arteries disease; CABG—coronary artery bypass graft surgery; CT—computer tomography; hs-cTnI—high sensitivity cardiac troponin-I; LAD—left anterior descending coronary artery; LCX—left circumflex artery; LMS—left main stem; NSTEMI—non-ST segment elevation myocardial infarction; PCI—percutaneous coronary intervention; RCA—right coronary artery; UA—unstable angina. ^a^—There is a significant difference between sexes in all patients (*p*-value <0.001). ^b^—There is no significant difference between sexes in MI1 group (*p*-value 0.19). ^c^—There is a significant difference between sexes in UA group (*p*-value 0.005). Bolded *p* values are statistically significant, names of variables groups are bolded.

**Table 2 medicina-58-01084-t002:** Median values of high sensitivity cardiac troponin-I after reclassification.

	All Patients	UA	Type 1 NSTEMI	MINOCA	Probable Type 1 NSTEMI	Probable MINOCA	Elevated after Coronary Angiography
	Median (IQR)	Median (IQR)	Median (IQR)	Median (IQR)	Median (IQR)	Median (IQR)	Median (IQR)
**Hs-cTnI at the presentation, ** **ng/L**	17.6 (902.05)	6.4 (8.10)	184.4 (226.15)	184.7 (165.15)	47.2 (67.25)	47.0 (28.45)	6.9 (7.30)
**Lowest hs-cTnI, ** **ng/L**	51.7 (125.99)	5.7 (7.28)	114.0 (207.40)	125.6 (147.30)	37.0 (50.80)	38.7 (33.08)	6.9 (3.10)
**Highest hs-cTnI, ** **ng/L**	115.2 (1112.71)	7.1 (8.80)	304.0 (357.60)	200.3 (317.05)	61.2 (55.20)	49.5 (29.15)	1522.0 (2069.30)

IQR—interquartile range; hs-cTnI—high sensitivity cardiac troponin-I; MINOCA—myocardial infarction with non-obstructive coronary arteries; NSTEMI—non-ST segment elevation myocardial infarction, UA—unstable angina.

## Data Availability

The data that support the findings of this study are available from the corresponding author upon reasonable request.

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
