# Peer review of "High-Sensitivity Cardiac Troponin Impact on the Differential Diagnosis of Non-ST Segment Elevation Coronary Syndromes—Is It Helping?"

_medicina, 2022, doi:10.3390/medicina58081084_

Round 1
Reviewer 1 Report
The authors tried to perform a reclassification of patients admitted with acute coronary syndomes without ST segment elevation on ECG based on hs-cTnI levels. It is of interest to analyse how we, health care professionals apply the guidelines on clinical practice.
However, there are some issues to be addressed by the authors.
Major.
1. Please give arguments why did you choose > 5 times increase of the hs-cTnI as a threshold for NSTEMI diagnosis.
2. It is difficult to follow the analysis. In Table 1 there are 715 patients in total but in the group analysis are included 334 UA patients and 137 patients in NSTEMI. I suggest giving initially the information about all the patients before reclassification followed by the table after they have been reclassified. Please reorganize the flow-chart providing initially how the patients were classified in baseline and after the reclassification to better undestand the diffencence.
Minor
1. I suggest to replace in line 43 “doctors” with “ health care professionals”
2. Please devide legend of Figure 1 from the text bellow.
3. Please specify “increasing angina”. Do you mean destabilization of previously stable angina or crescendo angina?
4. Why to chose 5 fold increase threshold
5. Please organize better table 1: i.e put baseline characteristics on the left and so on to give order to the patient characteristics overall.
6. It is difficult to differentiate the coded colors in the legend of the Figure 2. You may increase the square size or change the color.
Author Response
Hello, our team wants to thank you for the review and the comments we received. It was really helpful.
Here are our comments on the mentioned issues.
- For the threshold we chose > 5 times increase of the hs-cTnI. It is because eleva-tions beyond 5-fold the upper reference limit have high (>90%) positive predictive value (PPV) for acute type 1 myocardial infarction (MI) whereas elevations up to 3-fold the upper reference limit have only limited (50–60%) PPV for acute MI and may be associated with a broad spectrum of conditions (e.g. tachyarrhythmias, myocarditis, heart failure)
- We changed the style of the flowchart and added some additional information about reclassification process, why we analysed UA and NSTEMI groups in the manuscript.
- All minor suggestions were adressed.
Reviewer 2 Report
I read with interest the article entitled „ High Sensitivity Cardiac Troponin Impact on non-ST
Elevation Coronary Syndromes Differential Diagnosis - is it
helping?” which focuses on an important clinically issue- “borderline” patients UA/NSTEMI “. The manuscript contains valuable comments and information and is generally well-written. However, I have several comments regarding this manuscript.
1. My general comments regard the local nature of the data collected (Lithuania). In my opinion, the treatment of patients presenting with chest pain and/or suspected diagnosis of UA depends largely on the organization of the health service, the availability of invasive diagnostics, financing of these procedures, and the local protocol of management. I think authors should comment on this fact in their manuscript.
2. The patients included in the study were hospitalized or only provided with emergency care within the ER. If they were hospitalized, whether they were admitted to the cardiology department or other departments? (It would be interesting to see the characteristics of the data regarding this issue)
3 Material and Methods section- please clarify which departments of Vilnius University Hospital were included to the analysis (particular discharging department is interesting
4. Excluded patients with "probable" type 1 NSTEMI and MINOC with intermediate increase hs TnI – in my opinion, is the most interesting in this study. Authors should comment on their status and I hope that they will try to analyze this population in the next manuscript
5 I am truly surprised by the fact that almost every fifth patient from the UA group had NSTEMI – did the original diagnosis was set up by a trained cardiologist??
6. “15 percent of patients with discharge UA diagnosis had no hs-cTnI assay or CAG performed “ please make a comment on this fact. In my everyday practice discharging a patient with a diagnosis of UA without coronary angiography or at least a non-invasive assessment of myocardial ischemia from the hospital with a Cathlab is an extremely rare situation. Unstable means potentially life-threatening- what has happened with the population of patients? Did they have scheduled CA or any kind of ischemia assessment?
7 Additional limitation section would be advisable.
8 What are the general numbers of ACS patients treated in this particular hospital – 840 patients diagnosed with UA during 2 years of follow-up in the single-center study it is quite a lot.. What are the numbers for STEMI/NSTEMI/ stable angina - PCI…??
8. Congratulations on the work done during the preparation of this manuscript - the detailed re-analysis of medical history, ECG records, and laboratory tests of such a high number of patients required a lot of effort.
Author Response
Hello, our team wants to thank you for the review and the comments we received. It was really helpful.
Here are our comments on the mentioned issues.
- Vilnius University Hospital Santaros Klinikos Centre of Cardiology and Angiology is a tertiary care center. A broad specter of procedures is available for managing patients with acute coronary syndromes. All procedures are performed 24 hours per day, 7 days per week and are free of charge for the patients since we have universal health care provided by the state. (We added this information in the manuscript as well)
-
and 3. All the patients were admitted to one of the departments of Vilnius University Hospital Centre of Cardiology and Angiology (1st and 2nd Department of Cardiology, Department of arrhythmias, Department of Interventional Cardiology and Cardiovascular Intensive care unit). Patients were discharged from one of the departments mentions earlier except Cardiology and Cardiovascular Intensive care unit.
4. After reclassification, we did not include MINOCA and “probable” MINOCA groups in the final statistical analysis due to the small sample. For final statistical analysis, we chose to compare the UA and NSTEMI groups because of a large sample, and these groups best met reclassification criteria and there is no need for further investigations for these diagnoses to be confirmed
5. Yes all the patients were treated by trained cardiologist.
6. For patients in our study CAG wasn‘t performed because:
- patients didn’t gave the consent for CAG;
- there were no coronary signs of myocardial ischemia in non-invasive exercise tolerance test;
- CAG and PCI was performed recently;
- triple vessel disease was found and CAG at that moment was too risky;
- patient died while waiting for the procedure
- hs-TnI was performed in other center, patient was transfered with UA diagnosis and unfortunately the value of the assay was not mentioned in clinical history.
We included this information in the manuscript as well.
7. This study has four main limitations. First, the study is held only in one center although there are six PCI centers in Lithuania and each has its own practice and algorithms for diagnosing and managing UA and the type 1 NSTEMI. Second, we did not analyze MINOCA, or probable MINOCA groups because of the small number of patients and the need for further investigations for these diagnoses to be confirmed. Third is that anamnesis data collected (i. e. chest pain characteristics, smoking anamnesis) is not very accurate because our center does not have one questionnaire for patients that present to ER with chest pain and data differs between each specialist. The last one is that we have not performed a follow-up of the UA and the type 1 NSTEMI groups patients yet.
We included this information in the manuscript as well.
8. Vilnius University Hospital Santaros Klinikos Centre of Cardiology and Angiology is a tertiary hospital centre that yearly performs about 2500 PCI (700 out of them are pri-mary PCI) and about 900 cases of MI are hospitalised. (This information is also added to the manuscript).
Round 2
Reviewer 2 Report
The authors responded to all my comments. The manuscript has improved. I have no new comments.